# On Lazy Training in Differentiable Programming

**Lénaïc Chizat**
CNRS, Université Paris-Sud
Orsay, France
lenaic.chizat@u-psud.fr

**Edouard Oyallon**
CentraleSupelec, INRIA
Gif-sur-Yvette, France
edouard.oyallon@centralesupelec.fr

**Francis Bach**
INRIA, ENS, PSL Research University
Paris, France
francis.bach@inria.fr

## Abstract

In a series of recent theoretical works, it was shown that strongly over-parameterized neural networks trained with gradient-based methods could converge exponentially fast to zero training loss, with their parameters hardly varying. In this work, we show that this "lazy training" phenomenon is not specific to over-parameterized neural networks, and is due to a choice of scaling, often implicit, that makes the model behave as its linearization around the initialization, thus yielding a model equivalent to learning with positive-definite kernels. Through a theoretical analysis, we exhibit various situations where this phenomenon arises in non-convex optimization and we provide bounds on the distance between the lazy and linearized optimization paths. Our numerical experiments bring a critical note, as we observe that the performance of commonly used non-linear deep convolutional neural networks in computer vision degrades when trained in the lazy regime. This makes it unlikely that "lazy training" is behind the many successes of neural networks in difficult high dimensional tasks.

## 1 Introduction

Differentiable programming is becoming an important paradigm in signal processing and machine learning that consists in building parameterized models, sometimes with a complex architecture and a large number of parameters, and adjusting these parameters in order to minimize a loss function using gradient-based optimization methods. The resulting problem is in general highly non-convex. It has been observed empirically that, for fixed loss and model class, changes in the parameterization, optimization procedure, or initialization could lead to a selection of models with very different properties [36]. This paper is about one such implicit bias phenomenon, that we call *lazy training*, which corresponds to the model behaving like its linearization around the initialization.

This work is motivated by a series of recent articles [11, 22, 10, 2, 37] where it is shown that over-parameterized neural networks could converge linearly to zero training loss with their parameters hardly varying. With a slightly different approach, it was shown in [17] that infinitely wide neural networks behave like the linearization of the neural network around its initialization. In the present work, we argue that this behavior is not specific to neural networks, and is not so much due to over-parameterization than to an implicit choice of scaling. By introducing an explicit scale factor, we show that essentially any parametric model can be trained in this lazy regime if its output is close to zero at initialization. This shows that guaranteed fast training is indeed often possible, but

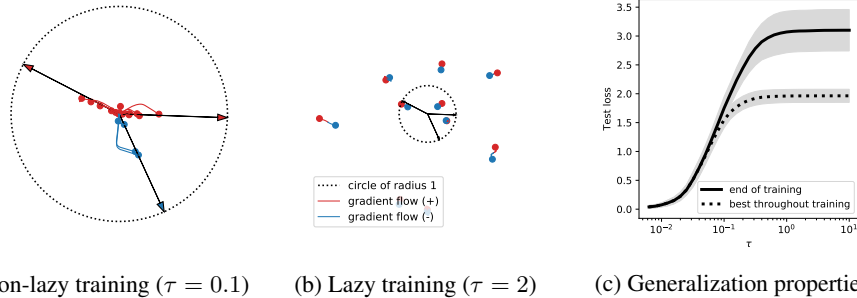

(a) Non-lazy training ($\tau = 0.1$)    (b) Lazy training ($\tau = 2$)    (c) Generalization properties

Figure 1: Training a two-layer ReLU neural network initialized with normal random weights of variance $\tau^2$: lazy training occurs when $\tau$ is large. (a)-(b) Trajectory of weights during gradient descent in 2-D (color shows sign of output layer). (c) Generalization in 100-D: it worsens as $\tau$ increases. The ground truth is generated with 3 neurons (arrows in (a)-(b)). Details in Section 3.

at the cost of recovering a linear method[1]. Our experiments on two-layer neural networks and deep convolutional neural networks (CNNs) suggest that this behavior is undesirable in practice.

## 1.1 Presentation of lazy training

We consider a parameter space[2] $\mathbb{R}^p$, a Hilbert space $\mathcal{F}$, a smooth model $h : \mathbb{R}^p \to \mathcal{F}$ (such as a neural network) and a smooth loss $R : \mathcal{F} \to \mathbb{R}_+$. We aim to minimize, with gradient-based methods, the objective function $F : \mathbb{R}^p \to \mathbb{R}_+$ defined as

$$F(w) \coloneqq R(h(w)).$$

With an initialization $w_0 \in \mathbb{R}^p$, we define the linearized model $\bar{h}(w) = h(w_0) + Dh(w_0)(w - w_0)$ around $w_0$, and the corresponding objective $\bar{F} : \mathbb{R}^p \to \mathbb{R}_+$ as

$$\bar{F}(w) \coloneqq R(\bar{h}(w)).$$

It is a general fact that the optimization path of $F$ and $\bar{F}$ starting from $w_0$ are close at the beginning of training. We call *lazy training* the less expected situation where these two paths remain close until the algorithm is stopped.

Showing that a certain non-convex optimization is in the lazy regime opens the way for surprisingly precise results, because linear models are rather well understood. For instance, when $R$ is strongly convex, gradient descent on $\bar{F}$ with an appropriate step-size converges linearly to a global minimizer [4]. For two-layer neural networks, we show in Appendix A.2 that the linearized model is a random feature model [26] which lends itself nicely to statistical analysis [6]. Yet, while advantageous from a theoretical perspective, it is not clear *a priori* whether this lazy regime is desirable in practice.

This phenomenon is illustrated in Figure 1 where lazy training for a two-layer neural network with rectified linear units (ReLU) is achieved by increasing the variance $\tau^2$ at initialization (see next section). While in panel (a) the ground truth features are identified, this is not the case for lazy training on panel (b) that manages to interpolate the observations with just a small displacement in parameter space (in both cases, near zero training loss was achieved). As seen on panel (c), this behavior hinders good generalization in the teacher-student setting [30]. The plateau reached for large $\tau$ corresponds exactly to the performance of the linearized model, see Section 3.1 for details.

## 1.2 When does lazy training occur?

**A general criterion.** Let us start with a formal computation. We assume that $w_0$ is not a minimizer so that $F(w_0) > 0$, and not a critical point so that $\nabla F(w_0) \neq 0$. Consider a gradient descent step $w_1 \coloneqq w_0 - \eta \nabla F(w_0)$, with a small stepsize $\eta > 0$. On the one hand, the relative change of the objective is $\Delta(F) \coloneqq \frac{|F(w_1) - F(w_0)|}{F(w_0)} \approx \eta \frac{\|\nabla F(w_0)\|^2}{F(w_0)}$. On the other hand, the relative change of the

differential of $h$ measured in operator norm is $\Delta(Dh) := \frac{\|Dh(w_1)-Dh(w_0)\|}{\|Dh(w_0)\|} \leq \eta \frac{\|\nabla F(w_0)\| \cdot \|D^2 h(w_0)\|}{\|Dh(w_0)\|}$. Lazy training refers to the case where the differential of $h$ does not sensibly change while the loss enjoys a significant decrease, i.e., $\Delta(F) \gg \Delta(Dh)$. Using the above estimates, this is guaranteed when

$$\frac{\|\nabla F(w_0)\|}{F(w_0)} \gg \frac{\|D^2 h(w_0)\|}{\|Dh(w_0)\|}.$$

For the square loss $R(y) = \frac{1}{2}\|y - y^\star\|^2$ for some $y^\star \in \mathcal{F}$, this leads to the simpler criterion

$$\kappa_h(w_0) := \|h(w_0) - y^\star\| \frac{\|D^2 h(w_0)\|}{\|Dh(w_0)\|^2} \ll 1, \tag{1}$$

using the approximation $\|\nabla F(w_0)\| = \|Dh(w_0)^\intercal (h(w_0) - y^\star)\| \approx \|Dh(w_0)\| \cdot \|h(w_0) - y^\star\|$. This quantity $\kappa_h(w_0)$ could be called the inverse *relative scale* of the model $h$ at $w_0$. We prove in Theorem 2.3 that it indeed controls how much the training dynamics differs from the linearized training dynamics when $R$ is the square loss[3]. For now, let us explore situations in which lazy training can be shown to occur, by investigating the behavior of $\kappa_h(w_0)$.

**Rescaled models.** Considering a scaling factor $\alpha > 0$, it holds

$$\kappa_{\alpha h}(w_0) = \frac{1}{\alpha}\|\alpha h(w_0) - y^\star\| \frac{\|D^2 h(w_0)\|}{\|Dh(w_0)\|^2}.$$

Thus, $\kappa_{\alpha h}(w_0)$ simply decreases as $\alpha^{-1}$ when $\alpha$ grows and $\|\alpha h(w_0) - y^\star\|$ is bounded, leading to lazy training for large $\alpha$. Training dynamics for such rescaled models are studied in depth in Section 2. For neural networks, there are various ways to ensure $h(w_0) = 0$, see Section 3.

**Homogeneous models.** If $h$ is $q$-positively homogeneous[4] then multiplying the initialization by $\lambda$ is equivalent to multiplying the scale factor $\alpha$ by $\lambda^q$. In equation,

$$\kappa_h(\lambda w_0) = \frac{1}{\lambda^q}\|\lambda^q h(w_0) - y^\star\| \frac{\|D^2 h(w_0)\|}{\|Dh(w_0)\|^2}.$$

This formula applies for instance to $q$-layer neural networks consisting of a cascade of homogenous non-linearities and linear, but not affine, operators. Such networks thus enter the lazy regime as the variance of initialization increases, if one makes sure that the initial output has bounded norm (see Figures 1 and 2(b) for 2-homogeneous examples).

**Two-layer neural networks.** For $m, d \in \mathbb{N}$, consider functions $h_m : (\mathbb{R}^d)^m \to \mathcal{F}$ of the form

$$h_m(w) = \alpha(m) \sum_{i=1}^{m} \phi(\theta_i),$$

where $\alpha(m) > 0$ is a normalization, $w = (\theta_1, \ldots, \theta_m)$ and $\phi : \mathbb{R}^d \to \mathcal{F}$ is a smooth function. This setting covers the case of two-layer neural networks (see Appendix A.2). When initializing with independent and identically distributed variables $(\theta_i)_{i=1}^m$ satisfying $\mathbb{E}\phi(\theta_i) = 0$, and under the assumption that $D\phi$ is not identically 0 on the support of the initialization, we prove in Appendix A.2 that for large $m$ it holds

$$\mathbb{E}[\kappa_{h_m}(w_0)] \lesssim m^{-\frac{1}{2}} + (m\alpha(m))^{-1}.$$

As a consequence, as long as $m\alpha(m) \to \infty$ when $m \to \infty$, such models are bound to reach the lazy regime. In this case, the norm of the initial output becomes negligible in front of the scale as $m$ grows due to the statistical cancellations that follow from the assumption $\mathbb{E}\phi(\theta_i) = 0$. In contrast, the critical scaling $\alpha(m) = 1/m$, allows to converge as $m \to \infty$ to a non degenerate dynamic described by a partial differential equation and referred to as the mean-field limit [24, 7, 29, 33].

## 1.3 Content and contributions

The goal of this paper is twofold: (i) understanding in a general optimization setting when lazy training occurs, and (ii) investigating the practical usefulness of models in the lazy regime. It is organized as follows:

- in Section 2, we study the gradient flows for rescaled models $\alpha h$ and prove in various situations that for large $\alpha$, they are close to gradient flows of the linearized model. When the loss is strongly convex, we also prove that lazy gradient flows converge linearly, either to a global minimizer for over-parameterized models, or to a local minimizer for under-parameterized models.

- in Section 3, we use numerical experiments on synthetic cases to illustrate how lazy training differs from other regimes of training (see also Figure 1). Most importantly, we show empirically that CNNs used in practice could be far from the lazy regime, with their performance not exceeding that of some classical linear methods as they become lazy.

Our focus is on general principles and qualitative description.

**Updates of the paper.** This article is an expanded version of "A Note on Lazy Training in Supervised Differential Programming" that appeared online in December 2018. Compared to the first version, it has been complemented with finite horizon bounds in Section 2.2 and numerical experiments on CNNs in Section 3.2 while the rest of the material has been slightly reorganized.

## 2 Analysis of Lazy Training Dynamics

### 2.1 Theoretical setting

Our goal in this section is to show that lazy training dynamics for the scaled objective

$$F_\alpha(w) := \frac{1}{\alpha^2} R(\alpha h(w)) \tag{2}$$

are close, when the scaling factor $\alpha$ is large, to those of the scaled objective for the linearized model

$$\bar{F}_\alpha(w) := \frac{1}{\alpha^2} R(\alpha \bar{h}(w)), \tag{3}$$

where $\bar{h}(w) := h(w_0) + Dh(w_0)(w - w_0)$ and $w_0 \in \mathbb{R}^p$ is a fixed initialization. Multiplying the objective by $1/\alpha^2$ does not change the minimizers, and corresponds to the proper time parameterization of the dynamics for large $\alpha$. Our basic assumptions are the following:

**Assumption 2.1.** *The parametric model $h : \mathbb{R}^p \to \mathcal{F}$ is differentiable with a locally Lipschitz differential[5] $Dh$. Moreover, $R$ is differentiable with a Lipschitz gradient.*

This setting is mostly motivated by supervised learning problems, where one considers a probability distribution $\rho \in \mathcal{P}(\mathbb{R}^d \times \mathbb{R}^k)$ and defines $\mathcal{F}$ as the space $L^2(\rho_x; \mathbb{R}^k)$ of square-integrable functions with respect to $\rho_x$, the marginal of $\rho$ on $\mathbb{R}^d$. The risk $R$ is then built from a smooth loss function $\ell : (\mathbb{R}^k)^2 \to \mathbb{R}_+$ as $R(g) = \mathbb{E}_{(X,Y)\sim\rho}\ell(g(X), Y)$. This corresponds to empirical risk minimization when $\rho$ is a finite discrete measure, and to population risk minimization otherwise (in which case only stochastic gradients are available to algorithms). Finally, one defines $h(w) = f(w, \cdot)$ where $f : \mathbb{R}^p \times \mathbb{R}^d \to \mathbb{R}^k$ is a parametric model, such as a neural network, which outputs in $\mathbb{R}^k$ depend on parameters in $\mathbb{R}^p$ and input data in $\mathbb{R}^d$.

**Gradient flows.** In the rest of this section, we study the *gradient flow* of the objective function $F_\alpha$ which is an approximation of (accelerated) gradient descent [12, 31] and stochastic gradient descent [19, Thm. 2.1] with small enough step sizes. With an initialization $w_0 \in \mathbb{R}^p$, the gradient

flow of $F_\alpha$ is the path $(w_\alpha(t))_{t\geq 0}$ in the space of parameters $\mathbb{R}^p$ that satisfies $w_\alpha(0) = w_0$ and solves the ordinary differential equation

$$w'_\alpha(t) = -\nabla F_\alpha(w_\alpha(t)) = -\frac{1}{\alpha}Dh(w_\alpha(t))^\intercal \nabla R(\alpha h(w_\alpha(t))), \tag{4}$$

where $Dh^\intercal$ denotes the adjoint of the differential $Dh$. We will study this dynamic for itself, and will also compare it to the gradient flow $(\bar{w}_\alpha(t))_{t\geq 0}$ of $\bar{F}_\alpha$ that satisfies $\bar{w}_\alpha(0) = w_0$ and solves

$$\bar{w}'_\alpha(t) = -\nabla \bar{F}_\alpha(\bar{w}_\alpha(t)) = -\frac{1}{\alpha}Dh(w_0)^\intercal \nabla R(\alpha \bar{h}(\bar{w}_\alpha(t))). \tag{5}$$

Note that when $h(w_0) = 0$, the renormalized dynamic $w_0 + \alpha(\bar{w}_\alpha(t) - w_0)$ does not depend on $\alpha$, as it simply follows the gradient flow of $w \mapsto R(Dh(w_0)(w - w_0))$ starting from $w_0$.

## 2.2 Bounds with a finite time horizon

We start with a general result that confirms that when $h(w_0) = 0$, taking large $\alpha$ leads to lazy training. We do not assume convexity of $R$.

**Theorem 2.2** (General lazy training). *Assume that $h(w_0) = 0$. Given a fixed time horizon $T > 0$, it holds $\sup_{t\in[0,T]} \|w_\alpha(t) - w_0\| = O(1/\alpha)$,*

$$\sup_{t\in[0,T]} \|w_\alpha(t) - \bar{w}_\alpha(t)\| = O(1/\alpha^2) \quad \text{and} \quad \sup_{t\in[0,T]} \|\alpha h(w_\alpha(t)) - \alpha\bar{h}(\bar{w}_\alpha(t))\| = O(1/\alpha).$$

For supervised machine learning problems, the bound on $\|w_\alpha(t) - \bar{w}_\alpha(t)\|$ implies that $\alpha h(w_\alpha(T))$ also *generalizes* like $\alpha\bar{h}(\bar{w}_\alpha(T))$ outside of the training set for large $\alpha$, see Appendix A.3. Note that the generalization behavior of linear models has been widely studied, and is particularly well understood for random feature models [26], which are recovered when linearizing two layer neural networks, see Appendix A.2. It is possible to track the constants in Theorem 2.2 but they would depend exponentially on the time horizon $T$. This exponential dependence can however be discarded for the specific case of the square loss, where we recover the *scale* criterion informally derived in Section 1.2.

**Theorem 2.3** (Square loss, quantitative). *Consider the square loss $R(y) = \frac{1}{2}\|y - y^\star\|^2$ for some $y^\star \in \mathcal{F}$ and assume that for some (potentially small) $r > 0$, $h$ is $\mathrm{Lip}(h)$-Lipschitz and $Dh$ is $\mathrm{Lip}(Dh)$-Lipschitz on the ball of radius $r$ around $w_0$. Then for an iteration number $K > 0$ and corresponding time $T := K/\mathrm{Lip}(h)^2$, it holds*

$$\frac{\|\alpha h(w_\alpha(T)) - \alpha\bar{h}(\bar{w}_\alpha(T))\|}{\|\alpha h(w_0) - y^\star\|} \leq \frac{K^2}{\alpha}\frac{\mathrm{Lip}(Dh)}{\mathrm{Lip}(h)^2}\|\alpha h(w_0) - y^\star\|$$

*as long as $\alpha \geq K\|\alpha h(w_0) - y^\star\|/(r\mathrm{Lip}(h))$.*

We can make the following observations:

- For the sake of interpretability, we have introduced a quantity $K$, analogous to an iteration number, that accounts for the fact that the gradient flow needs to be integrated with a step-size of order $1/\mathrm{Lip}(\nabla F_\alpha) = 1/\mathrm{Lip}(h)^2$. For instance, with this step-size, gradient descent at iteration $K$ approximates the gradient flow at time $T = K/\mathrm{Lip}(h)^2$, see, e.g., [12, 31].
- Laziness only depends on the local properties of $h$ around $w_0$. These properties may vary a lot over the parameter space, as is the case for homogeneous functions seen in Section 1.2.

For completeness, similar bounds on $\|w_\alpha(T) - w_0\|$ and $\|w_\alpha(T) - \bar{w}_\alpha(T)\|$ are also provided in Appendix B.2. The drawback of the bounds in this section is the increasing dependency in time, which is removed in the next section. Yet, the relevance of Theorem 2.2 remains because it does not depend on the conditioning of the problem. Although the bound grows as $K^2$, it gives an informative estimate for large or ill-conditioned problems, where training is typically stopped much before convergence.

## 2.3 Uniform bounds and convergence in the lazy regime

This section is devoted to uniform bounds in time and convergence results under the assumption that $R$ is strongly convex. In this setting, the function $\bar{F}_\alpha$ is strictly convex on the affine hyperspace

$w_0 + \ker Dh(w_0)^\perp$ which contains the linearized gradient flow $(\bar{w}_\alpha(t))_{t \geq 0}$, so the latter converges linearly to the unique global minimizer of $\bar{F}_\alpha$. In particular, if $h(w_0) = 0$ then this global minimizer does not depend on $\alpha$ and $\sup_{t \geq 0} \|\bar{w}_\alpha(t) - w_0\| = O(1/\alpha)$. We will see in this part how these properties reflect on the lazy gradient flow $w_\alpha(t)$.

**Over-parameterized case.** The following proposition shows global convergence of lazy training under the condition that $Dh(w_0)$ is surjective. As $\operatorname{rank} Dh(w_0)$ gives the number of effective parameters or degrees of freedom of the model around $w_0$, this over-parameterization assumption guarantees that any model around $h(w_0)$ can be fitted. Of course, this can only happen if $\mathcal{F}$ is finite-dimensional.

**Theorem 2.4** (Over-parameterized lazy training). *Consider a $M$-smooth and $m$-strongly convex loss $R$ with minimizer $y^\star$ and condition number $\kappa := M/m$. Assume that $\sigma_{\min}$, the smallest singular value of $Dh(w_0)^\intercal$ is positive and that the initialization satisfies $\|h(w_0)\| \leq C_0 := \sigma_{\min}^3/(32\kappa^{3/2}\|Dh(w_0)\|\operatorname{Lip}(Dh))$ where $\operatorname{Lip}(Dh)$ is the Lipschitz constant of $Dh$. If $\alpha > \|y^*\|/C_0$, then for $t \geq 0$, it holds*

$$\|\alpha h(w_\alpha(t)) - y^*\| \leq \sqrt{\kappa}\|\alpha h(w_0) - y^*\|\exp(-m\sigma_{\min}^2 t/4).$$

*If moreover $h(w_0) = 0$, it holds as $\alpha \to \infty$, $\sup_{t \geq 0} \|w_\alpha(t) - w_0\| = O(1/\alpha)$,*

$$\sup_{t \geq 0} \|\alpha h(w_\alpha(t)) - \alpha \bar{h}(\bar{w}_\alpha(t))\| = O(1/\alpha) \quad and \quad \sup_{t \geq 0} \|w_\alpha(t) - \bar{w}_\alpha(t)\| = O(\log \alpha/\alpha^2).$$

The proof of this result relies on the fact that $\alpha h(w_\alpha(t))$ follows the gradient flow of $R$ in a time-dependent and non degenerate metric: the pushforward metric [21] induced by $h$ on $\mathcal{F}$. For the first part, we do not claim improvements over [11, 22, 10, 2, 37], where a lot of effort is also put in dealing with the non-smoothness of $h$, which we do not study here. As for the uniform in time comparison with the tangent gradient flow, it is new and follows mostly from Lemma B.2 in Appendix B where the constants are given and depend polynomially on the characteristics of the problem.

**Under-parameterized case.** We now remove the over-parameterization assumption and show again linear convergence for large values of $\alpha$. This covers in particular the case of population loss minimization, where $\mathcal{F}$ is infinite-dimensional. For this setting, we limit ourselves to a qualitative statement[6].

**Theorem 2.5** (Under-parameterized lazy training). *Assume that $\mathcal{F}$ is separable, $R$ is strongly convex, $h(w_0) = 0$ and $\operatorname{rank} Dh(w)$ is constant on a neighborhood of $w_0$. Then there exists $\alpha_0 > 0$ such that for all $\alpha \geq \alpha_0$ the gradient flow (4) converges at a geometric rate (asymptotically independent of $\alpha$) to a local minimum of $F_\alpha$.*

Thanks to lower-semicontinuity of the rank function, the assumption that the rank is locally constant holds generically, in the sense that it is satisfied on an open dense subset of $\mathbb{R}^p$. In this under-parameterized case, the limit $\lim_{t \to \infty} w_\alpha(t)$ is for $\alpha$ large enough a strict local minimizer, but in general not a global minimizer of $F_\alpha$ because the image of $Dh(w_0)$ does not *a priori* contain the global minimizer of $R$. Thus it cannot be excluded that there exists parameters $w$ farther from $w_0$ with a smaller loss. This fact is clearly observed experimentally in Section 3, Figure 2-(b). Finally, a comparison with the linearized gradient flow as in Theorem 2.4 could be shown along the same lines, but would be technically slightly more involved because differential geometry comes into play.

**Relationship to the global convergence result in [7].** A consequence of Theorem 2.5 is that in the lazy regime, the gradient flow of the population risk for a two-layer neural network might get stuck in a local minimum. In contrast, it is shown in [7] that such gradient flows converge to global optimality in the infinite over-parameterization limit $p \to \infty$ if initialized with enough diversity in the weights. This is not a contradiction since Theorem 2.5 assumes a finite number $p$ of parameters. In the lazy regime, the population loss might also converge to its minimum when $p$ increases: this is guaranteed if the tangent kernel $Dh(w_0)Dh(w_0)^\intercal$ [17] converges (after normalization) to a universal kernel as $p \to \infty$. However, this convergence might be unreasonably slow in high-dimension, as Figure 1-(c) suggests. As a side note, we stress that the global convergence result in [7] is not limited to lazy dynamics but also covers non-linear dynamics, such as seen on Figure 1 where neurons move.

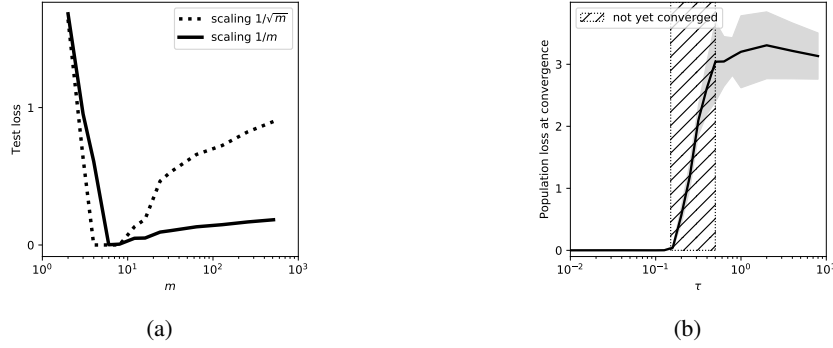

(a)                                         (b)

Figure 2: (a) Test loss at convergence for gradient descent, when $\alpha$ depends on $m$ as $\alpha = 1/m$ or $\alpha = 1/\sqrt{m}$, the latter leading to lazy training for large $m$ (not symmetrized). (b) Population loss at convergence versus $\tau$ for SGD with a random $\mathcal{N}(0, \tau^2)$ initialization (symmetrized). In the hatched area the loss was still slowly decreasing.

## 3 Numerical Experiments

We realized two sets of experiments, the first with two-layer neural networks conducted on synthetic data and the second with convolutional neural networks (CNNs) conducted on the CIFAR-10 dataset [18]. The code to reproduce these experiments is available online[7].

### 3.1 Two-layer neural networks in the teacher-student setting

We consider the following two-layer *student* neural network $h_m(w) = f_m(w, \cdot)$ with $f_m(w, x) = \sum_{j=1}^{m} a_j \max(b_j \cdot x, 0)$ where $a_j \in \mathbb{R}$ and $b_j \in \mathbb{R}^d$ for $j = 1, \ldots, m$. It is trained to minimize the square loss with respect to the output of a two-layer *teacher* neural network with same architecture and $m_0 = 3$ hidden neurons, with random weights normalized so that $\|a_j b_j\| = 1$ for $j \in \{1, 2, 3\}$. For the student network, we use random Gaussian weights, except when *symmetrized initialization* is mentioned, in which case we use random Gaussian weights for $j \leq m/2$ and set for $j > m/2$, $b_j = b_{j-m/2}$ and $a_j = -a_{j-m/2}$. This amounts to training a model of the form $h(w_a, w_b) = h_{m/2}(w_a) - h_{m/2}(w_b)$ with $w_a(0) = w_b(0)$ and guaranties zero output at initialization. The training data are $n$ input points uniformly sampled on the unit sphere in $\mathbb{R}^d$ and we minimize the empirical risk, except for Figure 2b(b) where we directly minimize the population risk with Stochastic Gradient Descent (SGD).

**Cover illustration.** Let us detail the setting of Figure 1 in Section 1. Panels (a)-(b) show gradient descent dynamics with $n = 15, m = 20$ with symmetrized initialization (illustrations with more neurons can be found in Appendix C). To obtain a 2-D representation, we plot $|a_j(t)|b_j(t)$ throughout training (lines) and at convergence (dots) for $j \in \{1, \ldots, m\}$. The blue or red colors stand for the signs of $a_j(t)$ and the unit circle is displayed to help visualizing the change of scale. On panel (c), we set $n = 1000, m = 50$ with symmetrized initialization and report the average and standard deviation of the test loss over 10 experiments. To ensure that the bad performances corresponding to large $\tau$ are not due to a lack of regularization, we display also the best test error throughout training (for kernel methods, early stopping is a form of regularization [34]).

**Increasing number of parameters.** Figure 2-(a) shows the evolution of the test error when increasing $m$ as discussed in Section 1.2, *without* symmetrized initialization. We report the results for two choices of scaling functions $\alpha(m)$, averaged over 5 experiments with $d = 100$. The scaling $1/\sqrt{m}$ leads to lazy training, with a poor generalization as $m$ increases, in contrast to the scaling $1/m$ for which the test error remains relatively close to 0 for large $m$ (more experiments with this scaling can be found in [7, 29, 24]).

**Under-parameterized case.**   Finally, Figure 2-(b) illustrates the under-parameterized case, with $d = 100, m = 50$ with symmetrized initialization. We used SGD with batch-size 200 to minimize the population square loss, and displayed average and standard deviation of the final population loss (estimated with 2000 samples) over 5 experiments. As shown in Theorem 2.5, SGD converges to a *a priori* local minimum in the lazy regime (i.e., here for large $\tau$). In contrast, it behaves well when $\tau$ is small, as in Figure 1. There is also an intermediate regime (hatched area) where convergence is very slow and the loss was still decreasing when the algorithm was stopped.

## 3.2   Deep CNNs experiments

We now study whether lazy training is relevant to understand the good performances of convolutional neural networks (CNNs).

**Interpolating from standard to lazy training.**   We first study the effect of increasing the scale factor $\alpha$ on a standard pipeline for image classification on the CIFAR10 dataset. We consider the VGG-11 model [32], which is a widely used model on CIFAR10. We trained it via mini-batch SGD with a momentum parameter of $0.9$. For the sake of interpretability, no extra regularization (e.g., BatchNorm) is incorporated, since a simple framework that outperforms linear methods baselines with some margin is sufficient to our purpose (see Figure 3(b)). An initial learning rate $\eta_0$ is linearly decayed at each epoch, following $\eta_t = \frac{\eta_0}{1+\beta t}$. The biases are initialized with 0 and all other weights are initialized with normal Xavier initialization [13]. In order to set the initial output to 0 we use the *centered model* $h$, which consists in replacing the VGG model $\tilde{h}$ by $h(w) := \tilde{h}(w) - \tilde{h}(w_0)$. Notice that this does not modify the differential at initialization.

The model $h$ is trained for the square loss multiplied by $1/\alpha^2$ (as in Section 2), with standard data-augmentation, batch-size of 128 [35] and $\eta_0 = 1$ which gives the best test accuracies over the grid $10^k$, $k \in \{-3, 3\}$, for all $\alpha$. The total number of epochs is 70, adjusted so that the performance reaches a plateau for $\alpha = 1$. Figure 3(a) reports the accuracy after training $\alpha h$ for increasing values of $\alpha \in 10^k$ for $k = \{0, 1, 2, 3, 4, 5, 6, 7\}$ ($\alpha = 1$ being the standard setting). For $\alpha < 1$, the training loss diverges with $\eta_0 = 1$. We also report the *stability of activations*, which is the share of neurons over ReLU layers that, after training, are activated for the same inputs than at initialization, see Appendix C. Values close to $100\%$ are strong indicators of an effective linearization.

We observe a significant drop in performance as $\alpha$ grows, and then the accuracy reaches a plateau, suggesting that the CNN progressively reaches the lazy regime. This demonstrates that the linearized model (large $\alpha$) is not sufficient to explain the good performance of the model for $\alpha = 1$. For large $\alpha$, we obtain a low limit training accuracy and do not observe overfitting, a surprising fact since this amounts to solving an over-parameterized linear system. This behavior is due to a poorly conditioned linearized model, see Appendix C.

**Performance of linearized CNNs.**   In this second set of experiments, we investigate whether variations of the models trained above in a lazy regime could increase the performance and, in particular, could outperform other linear methods which also do not involve learning a representation [26, 25]. To this end, we train widened CNNs in the lazy regime, as widening is a well-known strategy to boost performances of a given architecture [35]. We multiply the number of channels of each layer by 8 for the VGG model and 7 for the ResNet model [16] (these values are limited by hardware constraints). We choose $\alpha = 10^7$ to train the linearized models, a batch-size of 8 and, after cross-validation, $\eta_0 = 0.01, 1.0$ for respectively the standard and linearized model. We also multiply the initial weights by respectively 1.2 and 1.3 for the ResNet-18 and VGG-11, as we found that it slightly boosts the training accuracies. Each model is trained with the cross-entropy loss divided by $\alpha^2$ until the test accuracy stabilizes or increases, and we check that the average stability of activations (see Appendix C) was $100\%$.

As seen on Figure 3(b), widening the VGG model slightly improves the performances of the linearized model compared to the previous experiment but there is still a substantial gap of performances from other non-learned representations [28, 25] methods, not to mention the even wider gap with their non-lazy counterparts. This behavior is also observed on the state-of-the-art ResNet architecture. Note that [3] reports a test accuracy of $77.4\%$ without data augmentation for a linearized CNN with a specially designed architecture which in particular solves the issue of ill-conditioning. Whether

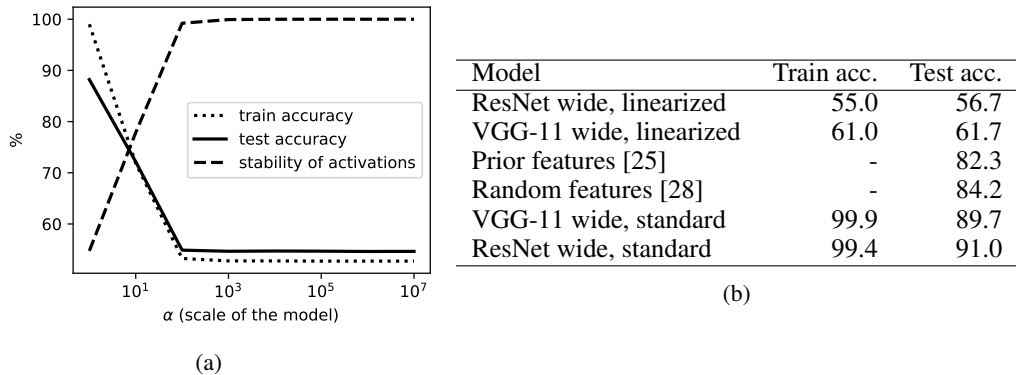

| Model | Train acc. | Test acc. |
|---|---|---|
| ResNet wide, linearized | 55.0 | 56.7 |
| VGG-11 wide, linearized | 61.0 | 61.7 |
| Prior features [25] | - | 82.3 |
| Random features [28] | - | 84.2 |
| VGG-11 wide, standard | 99.9 | 89.7 |
| ResNet wide, standard | 99.4 | 91.0 |

(b)

(a)

Figure 3: (a) Accuracies on CIFAR10 as a function of the scaling $\alpha$. The stability of activations suggest a linearized regime when high. (b) Accuracies on CIFAR10 obtained for $\alpha = 1$ (standard, non-linear) and $\alpha = 10^7$ (linearized) compared to those reported for some linear methods without data augmentation: random features and prior features based on the scattering transform.

variations of standard architectures and pipelines can lead to competitive performances with linearized CNNs, remains an open question.

**Remark on wide NNs.** It was proved [17] that neural networks with standard initialization (random independent weights with zero mean and variance $O(1/n_\ell)$ at layer $\ell$, where $n_\ell$ is the size of the previous layer), are bound to reach the lazy regime as the sizes of all layers grow unbounded. Moreover, for very large neural networks of more than 2 layers, this choice of initialization is essentially mandatory to avoid exploding or vanishing initial gradients [15, 14] if the weights are independent with zero mean. Thus we stress that we do not claim that wide neural networks do not show a lazy behavior, but rather that those which exhibit good performances are far from this asymptotic behavior.

## 4 Discussion

Lazy training is an implicit bias phenomenon, that refers to the situation when a non-linear parametric model behaves like a linear one. This arises when the *scale* of the model becomes large, which happens implicitly under some choices of hyper-parameters. While the lazy training regime provides some of the first optimization-related theoretical insights for deeper models [10, 2, 37, 17], we believe it does not explain yet the many successes of neural networks that have been observed in various challenging, high-dimensional tasks in machine learning. This is corroborated by numerical experiments where it is seen that the performance of networks trained in the lazy regime degrades and in particular does not exceed that of some classical linear methods. Instead, the intriguing phenomenon that still defies theoretical understanding is the one displayed on Figure 1(c) for small $\tau$ and on Figure 3(a) for $\alpha = 1$: neural networks trained with gradient-based methods (and neurons that move) have the ability to perform high-dimensional feature selection through highly non-linear dynamics.

### Acknowledgments

We acknowledge supports from grants from Région Ile-de-France and the European Research Council (grant SEQUOIA 724063). Edouard Oyallon was supported by a GPU donation from NVIDIA. We thank Alberto Bietti for interesting discussions and Brett Bernstein for noticing an error in a previous version of this paper.

## Footnotes

[1] Here we mean a prediction function linearly parameterized by a potentially infinite-dimensional vector.

[2] Our arguments could be generalized to the case where the parameter space is a Riemannian manifold.

[3]Note that lazy training could occur even when $\kappa_h(w_0)$ is large, i.e. Eq. (1) only gives a sufficient condition.

[4]That is, for $q \geq 1$, it holds $h(\lambda w) = \lambda^q h(w)$ for all $\lambda > 0$ and $w \in \mathbb{R}^p$.

[5] $Dh(w)$ is a continuous linear map from $\mathbb{R}^p$ to $\mathcal{F}$. The Lipschitz constant of $Dh : w \mapsto Dh(w)$ is defined with respect to the operator norm. When $\mathcal{F}$ has a finite dimension, $Dh(w)$ can be identified with the Jacobian matrix of $h$ at $w$.

[6]In contrast to the finite horizon bound of Theorem 2.3, quantitative statements would here involve the smallest positive singular value of $Dh(w_0)$, which is anyways hard to control.

[7]`https://github.com/edouardoyallon/lazy-training-CNN`

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
