[Supplementary Material]

## Supplementary material

Supplementary material for the paper: "On Lazy Training in Differentiable Programming" authored by Lénaïc Chizat , Edouard Oyallon and Francis Bach (NeurIPS 2019). This supplementary material is organized as follows:

- Appendix A: Remarks on the linearized model
- Appendix B: Proofs of the theoretical results
- Appendix C: Experimental details and additional results

## A  The linearized model in supervised machine learning

### A.1  Differentiable models and their linearization

In this section, we give some details on the interpretation of the linearized model in the case of supervised machine learning. In this setting, a differentiable model is a typically a function $f : \mathbb{R}^p \times \mathbb{R}^d \to \mathbb{R}^k$ where $\mathbb{R}^p$ is the parameter space, $\mathbb{R}^d$ is the input space and $\mathbb{R}^k$ the output space. One defines a Hilbert space $\mathcal{F}$ of functions from $\mathbb{R}^d$ to $\mathbb{R}^k$, typically $L^2(\rho_x, \mathbb{R}^k)$ where $\rho_x$ is the distribution of input samples. The function $h : \mathbb{R}^p \to \mathcal{F}$ considered in the article is then the function which to a vector of parameters associates a predictor $h : w \mapsto f(w, \cdot)$.

In first order approximation around the initial parameters $w_0 \in \mathbb{R}^p$, the parametric model $f(w, x)$ reduces to the following *linearized* or *tangent* model :

$$\bar{f}(w, x) = f(w_0, x) + D_w f(w_0, x)(w - w_0). \tag{6}$$

where $D_w f$ is the differential of $f$ in the variable $w$. The corresponding hypothesis class is *affine* in the space of predictors. It should be stressed that when $f$ is a neural network, $\bar{f}$ is generally not a linear neural network because it is not linear in $x \in \mathbb{R}^d$, but in the features $D_w f(w_0, x) \in \mathbb{R}^{p \times k}$ which generally depend non-linearly on $x$. For large neural networks, the dimension of the features might be much larger than $d$, which makes $\bar{f}$ similar to a non-parametric method. Finally, if $f$ is already a linear model, then $f$ and $\bar{f}$ are identical.

**Kernel method with an offset.**  In the case of the square loss, training the affine model (6) is equivalent to training a linear model in the variables

$$(\tilde{x}, \tilde{y}) := (D_w f(w_0, x), y - f(w_0, x)).$$

When $k = 1$, this is equivalent to a kernel method with the *tangent kernel* [17] defined as $K : \mathbb{R}^d \times \mathbb{R}^d \to \mathbb{R}$

$$K(x, x') = D_w f(w_0, x) D_w f(w_0, x')^\mathsf{T}. \tag{7}$$

This kernel is different from the one traditionally associated to neural networks [27, 9] which involve the derivative with respect to the output layer only. Also, the output data is shifted by the initialization of the model $h(w_0) = f(w_0, \cdot)$. This term inherits from the randomness due to the initialization: it is for instance shown in [20, 23] that the distribution of $h(w_0)$ converges to a Gaussian process for certain over-parameterized neural networks initialized with random normal weights.

### A.2  Two-layer neural networks

Lazy training has some interesting consequences when looking more particularly at two-layer neural networks. These are functions of the form

$$f_m(w, x) = \alpha(m) \sum_{j=1}^{m} b_j \cdot \sigma(a_j \cdot x),$$

where $m \in \mathbb{N}$ is the size of the hidden layer and $\sigma : \mathbb{R} \to \mathbb{R}$ is an activation function and the parameters[8] are $(\theta_j)_{j=1}^m$ where $\theta_j = (a_j, b_j) \in \mathbb{R}^{d+1}$, so here the number of parameters is $p = m(d+1)$. We have also introduced a scaling $\alpha(m) > 0$ as in Section 1.2.

**Justification for asymptotics.** In this paragraph, we justify the formula for the asymptotic upper bound on $\kappa_{h_m}(w_0)$ given for such models in Section 1.2. Using the assumption that $\mathbb{E}\phi(\theta_i) = 0$ and the fact that the parameters are independents, one has $\mathbb{E}\|h(w_0)\|^2 = m\alpha(m)^2\mathbb{E}\|\phi(\theta)\|^2$. For the differential, from the law of large numbers, we have the estimate

$$\frac{1}{m\alpha(m)^2}Dh(w_0)\,Dh(w_0)^\intercal = \frac{1}{m}\sum_{i=1}^m D\phi(\theta_i)D\phi(\theta_i)^\intercal \xrightarrow[m\to\infty]{} \mathbb{E}\left[D\phi(\theta)D\phi(\theta)^\intercal\right].$$

It follows that $\mathbb{E}\|Dh(w_0)\|^2 = \mathbb{E}\|Dh(w_0)Dh(w_0)^\intercal\| \sim m\alpha(m)^2\|\mathbb{E}[D\phi(\theta)D\phi(\theta)^\intercal]\|$ because we have assumed that $D\phi$ is not identically 0 on the support of $\theta$. One also has

$$\|D^2h(w_0)\| = \sup_{\substack{u\in\mathbb{R}^{d\times m} \\ \|u\|\le 1}} \alpha(m)\sum_{i=1}^m u_i^\intercal D^2\phi(\theta_i)u_i \le \alpha(m)\sup_{\theta_i}\|D^2\phi(\theta_i)\| \le \alpha(m)\mathrm{Lip}(D\phi).$$

From the definition of $\kappa_{h_m}(w_0)$ and the upper bound $\|h_m(w_0)-y^\star\| \le \|h(w_0)\|+\|y^\star\|$ we conclude that

$$\mathbb{E}[\kappa_{h_m}(w_0)] \lesssim m^{-\frac{1}{2}} + (m\alpha(m))^{-1}.$$

**Limit kernels and random feature.** In this section, we show that the tangent kernel is a *random feature kernel* for neural networks with a single hidden layer. For simplicity, we consider the scaling $\alpha(m) = 1/\sqrt{m}$ as in [11] which leads to a non-degenerated limit of the kernel[9] as $m \to \infty$. The associated tangent kernel in Eq. (7) is the sum of two kernels $K_m(x,x') = K_m^{(a)}(x,x') + K_m^{(b)}(x,x')$, one for each layer, where

$$K_m^{(a)}(x,x') = \frac{1}{m}\sum_{j=1}^m (x\cdot x')b_j^2\sigma'(a_j\cdot x)\sigma'(a_j\cdot x') \quad\text{and}\quad K_m^{(b)}(x,x') = \frac{1}{m}\sum_{j=1}^m \sigma(a_j\cdot x)\sigma(a_j\cdot x').$$

If we assume that the initial weights $a_j$ (resp. $b_j$) are independent samples of a distribution on $\mathbb{R}^d$ (resp. a distribution on $\mathbb{R}$), these are random feature kernels [26] that converge as $m \to \infty$ to the kernels

$$K^{(a)}(x,x') = \mathbb{E}_{(a,b)}\left[(x\cdot x')b^2\sigma'(a\cdot x)\sigma'(a\cdot x')\right] \quad\text{and}\quad K^{(b)}(x,x') = \mathbb{E}_a\left[\sigma(a\cdot x)\sigma(a\cdot x')\right].$$

The second component $K^{(b)}$, corresponding to the differential with respect to the output layer, is the one traditionally used to make the link between these networks and random features [27]. When $\sigma(s) = \max\{s,0\}$ is the rectified linear unit activation and the distribution of the weights $a_j$ is rotation invariant in $\mathbb{R}^d$, one has the following explicit formulae [8]:

$$K^{(a)}(x,x') = \frac{(x\cdot x')\mathbb{E}(b^2)}{2\pi}(\pi - \varphi), \quad K^{(b)}(x,x') = \frac{\|x\|\|x'\|\mathbb{E}(\|a\|^2)}{2\pi d}((\pi - \varphi)\cos\varphi + \sin\varphi) \tag{8}$$

where $\varphi \in [0,\pi]$ is the angle between the two vectors $x$ and $x'$. See Figure 4 for an illustration of this kernel and the convergence of its random approximations. The link with (independent) random sampling is lost for deeper neural networks, but it is shown in [17] that tangent kernels still converge when the size of networks increase, for certain architectures.

## A.3  Generalization for the lazy model

As noted in the main text, in supervised machine learning, $\mathcal{F}$ is often a Hilbert space of functions on $\mathbb{R}^d$ and the model $h$ is often of the form $h(w) = f(w,\cdot)$ where $f : \mathbb{R}^p \times \mathbb{R}^d \to \mathbb{R}^k$. A natural question that arises in this context and that is not directly answered by the theorems of Section 2, is whether the trained lazy model and the trained tangent model also generalize the same way, i.e. whether at training time $T$, it holds $f(w(T),x) \approx \bar{f}(\bar{w}(T),x)$ for points $x \in \mathbb{R}^d$ that are not in the training set, where $\bar{f}(w,x) = f(w_0,x) + D_w f(w_0,x)(w - w_0)$. We will see here that it is actually a simple consequence of the bounds.

Figure 4: Random realizations of the kernels $K_m$ and the limit kernel $K$ of Eq. (8). We display the value of $K(x, x')$ as a function of $\varphi = \text{angle}(x, x')$ with $x$ fixed, on a section of the sphere in $\mathbb{R}^{10}$. Parameters are normal random variables of variance 1, so $\mathbb{E}(b^2) = 1$ and $\mathbb{E}(\|a\|^2) = d$.

**Proposition A.1** (Generalizing like the tangent model). *Assume that for some $C > 0$ it holds $\|w_\alpha(T) - \bar{w}(T)\| \leq C \log(\alpha)/\alpha^2$. Assume moreover that there exists a set $\mathfrak{X} \subset \mathbb{R}^d$ such that $M_1 := \sup_{x \in \mathfrak{X}} \|D_w f(w_0, x)\| < \infty$ and $M_2 := \sup_{x \in \mathfrak{X}} \text{Lip}(w \mapsto D_w f(w, x)) < \infty$. Then it holds*

$$\sup_{x \in \mathfrak{X}} \|\alpha f(w_\alpha(T), x) - \alpha \bar{f}(\bar{w}_\alpha(T), x)\| \leq C \frac{\log \alpha}{\alpha} \left( M_1 + \frac{1}{2} C \cdot M_2 \cdot \log(\alpha) \right) \xrightarrow[\alpha \to \infty]{} 0.$$

*Proof.* Let us call $A$ the quantity to be upper bounded, and start with the decomposition

$$A \leq \sup_{x \in \mathfrak{X}} \|\alpha f(w_\alpha(T), x) - \alpha \bar{f}(w_\alpha(T), x)\| + \sup_{x \in \mathfrak{X}} \|\alpha \bar{f}(w_\alpha(T), x) - \alpha \bar{f}(\bar{w}_\alpha(T), x)\| = A_1 + A_2$$

By Taylor's theorem applied at each point $x \in X$, one has

$$A_1 \leq \frac{\alpha}{2} M_2 \|w_\alpha(T) - \bar{w}_\alpha(T)\|^2 \leq \frac{C^2 \cdot M_2 \log(\alpha)^2}{2\alpha}.$$

It also holds

$$A_2 = \alpha \sup_{x \in \mathfrak{X}} \|D_w f(w_0, x)(w_\alpha(T) - \bar{w}_\alpha(T))\| \leq \frac{M_1 C \log(\alpha)}{\alpha}$$

and the conclusion follows. $\qquad\square$

# B   Proofs of the theoretical results

In all the forthcoming proofs, we use the notations $y(t) = \alpha h(w_\alpha(t))$ and $\bar{y}(t) = \alpha \bar{h}(\bar{w}_\alpha(t))$ for the dynamics in $\mathcal{F}$ (they also depend on $\alpha$ although this is not reflected in the notation). We also write $\Sigma(w) := Dh(w)Dh(w)^\intercal$ for the so-called tangent kernel [17], which is a quadratic form on $\mathcal{F}$. By using the chain rule, we find that the trajectories in $\mathcal{F}$ solve the differential equation

$$\frac{d}{dt} y(t) = -\Sigma(w_\alpha(t)) \nabla R(y(t)),$$

$$\frac{d}{dt} \bar{y}(t) = -\Sigma(w(0)) \nabla R(\bar{y}(t)).$$

with $y(0) = \bar{y}(0) = \alpha h(w_0)$. Remark that the first differential equation is coupled with $w_\alpha(t)$.

## B.1   Proof for Theorem 2.2 (finite horizon, non-quantitative)

For this first proof, we only track the dependency in $\alpha$, and we use $C$ to denote a quantity independent of $\alpha$, that may vary from line to line. For $T > 0$, it holds

$$\int_0^T \|w'_\alpha(t)\| dt = \int_0^T \|\nabla F_\alpha(w_\alpha(t))\| dt \leq \sqrt{T} \left( \int_0^T \|\nabla F_\alpha(w_\alpha(t))\|^2 dt \right)^{\frac{1}{2}}.$$

It follows, by using the fact that $\frac{d}{dt}F_\alpha(w_\alpha(t)) = -\|\nabla F_\alpha(w_\alpha(t))\|^2$, that $\sup_{t\in[0,T]}\|w_\alpha(t) - w(0)\| \leq (T \cdot F_\alpha(w_\alpha(t)))^{\frac{1}{2}} \lesssim \frac{1}{\alpha}$. In particular, we deduce that $\sup_{t\in[0,T]}\|y(t) - y(0)\| \leq C$ and $\sup_{t\in[0,T]}\|\nabla R(y(t))\| \leq C$.

Let us now consider the evolution of $\Delta(t) := \|y(t) - \bar{y}(t)\|$. It satisfies $\Delta(0) = 0$ and

$$
\begin{aligned}
\Delta'(t) &\leq \|\Sigma(w_\alpha(t))\nabla R(y(t)) - \Sigma(w(0))\nabla R(\bar{y}(t))\| \\
&\leq \|(\Sigma(w_\alpha(t)) - \Sigma(w(0)))\nabla R(y(t))\| + \|\Sigma(w(0))(\nabla R(y(t)) - \nabla R(\bar{y}(t)))\| \\
&\leq C_1/\alpha + C_2\Delta(t)
\end{aligned}
$$

The ordinary differential equation $u'(t) = C_1/\alpha + C_2 u(t)$ with initial condition $u(0) = 0$ admits the unique solution $u(t) = \frac{C_1}{\alpha C_2}(\exp(C_2 t) - 1)$. Since $\Delta(t)$ is a sub-solution of this system, it follows that $\Delta(t) \leq \frac{C_1}{\alpha C_2}(\exp(C_2 t) - 1) \leq C/\alpha$ (notice the exponential dependence in the final time and some other characteristics of the problem). Finally, consider the quantity $\delta(t) = \|w_\alpha(t) - \bar{w}_\alpha(t)\|$. It holds

$$
\begin{aligned}
\delta'(t) &\leq \alpha^{-1}\|Dh(w_\alpha(t))^\intercal \nabla R(f(t)) - Dh(w_0)^\intercal \nabla R(\bar{y}(t))\| \\
&\leq \alpha^{-1}\|Dh(w_\alpha(t))^\intercal - Dh(w_0)^\intercal\|\|\nabla R(y(t))\| + \alpha^{-1}\|Dh(w_0)\|\|\nabla R(y) - \nabla R(\bar{y}(t))\| \\
&\leq C\alpha^{-2}
\end{aligned}
$$

We thus conclude, since $\delta(0) = 0$, that $\sup_{t\in[0,T]}\|\delta(t)\| \leq \alpha^{-2}$.

## B.2 Proof of Theorem 2.3 (finite horizon, square loss)

**Step 1.** With the square loss, the objective is still potentially non-convex, but we have the property

$$
\frac{d}{dt}\|y(t) - y^*\|^2 = -\langle \Sigma(w(t))(y(t) - y^\star), y(t) - y^\star\rangle \leq 0.
$$

The proof scheme is otherwise similar as above, but we carry all constants. Let us denote $T_{exit} = \inf\{t > 0 \; ; \; \|w_\alpha(t) - w_0\| > r\}$. For $t \leq T_{exit}$ it holds

$$
\|w'_\alpha(t)\| = \|\nabla F_\alpha(w_\alpha(t))\| \leq \alpha^{-1}\|y(t) - y^\star\|\|Dh(w_\alpha(t))\| \leq \alpha^{-1}\|y(0) - y^\star\|\mathrm{Lip}(h)
$$

It follows that $\|w_\alpha(t) - w(0)\| \leq t\alpha^{-1}\|y(0) - y^\star\|\mathrm{Lip}(h)$ (this bound is tighter for small times, compared to the bound in $\sqrt{t}$ used in the previous proof). Since we have assumed that $\alpha \geq k\|y(0) - y^\star\|/(r\mathrm{Lip}(h))$, it holds $\|w_\alpha(t) - w_0\| \leq (t/K) \cdot r\mathrm{Lip}(h)^2 = r$ so $T_{exit} > T$.

**Step 2.** Now we consider $\Delta(t) = \|y(t) - \bar{y}(t)\|$. It holds

$$
\begin{aligned}
\frac{1}{2}\frac{d}{dt}\Delta(t)^2 &= \langle y'(t) - \bar{y}'(t), y(t) - \bar{y}(t)\rangle \\
&\leq -\langle \Sigma(w_\alpha(t))\nabla R(y(t)) - \Sigma(w(0))\nabla R(\bar{y}(t)), y(t) - \bar{y}(t)\rangle \\
&\leq -\langle(\Sigma(w_\alpha(t)) - \Sigma(w(0)))\nabla R(y(t)), y(t) - \bar{y}(t)\rangle
\end{aligned}
$$

where we have used the fact that $\langle \Sigma(w(0))(\nabla R(y(t)) - \nabla R(\bar{y}(t))), y(t) - \bar{y}(t)\rangle \geq 0$, which is specific to the square loss. Taking the norms and dividing both sides by $\Delta(t)$, it follows

$$
\Delta'(t) \leq \mathrm{Lip}(\Sigma) \cdot \|w_\alpha(t) - w(0)\|\|y(0) - y^\star\| \leq 2\mathrm{Lip}(h)^2\mathrm{Lip}(Dh)t\alpha^{-1}\|y(0) - y^\star\|^2
$$

where we have used $\mathrm{Lip}(\Sigma) \leq 2\mathrm{Lip}(h)\mathrm{Lip}(Dh)$. Since $\Delta(0) = 0$, it follows

$$
\Delta(t) \leq \frac{t^2}{\alpha}\mathrm{Lip}(h)^2\mathrm{Lip}(Dh)\|y(0) - y^\star\|^2.
$$

The bound in the statement then follows by writing this upper bound at time $T = K/\mathrm{Lip}(h)^2$.

**Step 3.** Finally, consider $\delta(t) = \|w_\alpha(t) - \bar{w}_\alpha(t)\|$. The bound that we will obtain is not reported in the main text due to space constraints, but proved here for the sake of completeness. As in the previous proof, it holds

$$
\alpha\delta'(t) \leq \|Dh(w_\alpha(t))^\intercal - Dh(w_0)^\intercal\|\|\nabla R(y(t))\| + \|Dh(w_0)\|\|\nabla R(y) - \nabla R(\bar{y}(t))\| = A(t) + B(t).
$$

Let us bound these two quantities separately. On the one hand, it holds for $t \in [0, T]$,

$$A(t) \leq \mathrm{Lip}(Dh)\|w_\alpha(t) - w_0\|\|y(0) - y^\star\| \leq \frac{t}{\alpha}\mathrm{Lip}(h)\mathrm{Lip}(Dh)\|y(0) - y^\star\|^2.$$

On the other hand, it holds for $t \in [0, T]$,

$$B(t) \leq \frac{t^2}{\alpha}\mathrm{Lip}(h)^3\mathrm{Lip}(Dh)\|y(0) - y^\star\|^2.$$

By integrating these two bounds and summing, we get

$$\delta(T) \leq \frac{T^2}{\alpha^2}\mathrm{Lip}(h)^2\mathrm{Lip}(Dh)\|y(0) - y^\star\|^2 \left(\frac{2}{\mathrm{Lip}(h)} + \frac{4T}{3}\mathrm{Lip}(h)\right)$$

$$\leq \frac{K^2}{\alpha^2}\frac{\mathrm{Lip}(Dh)}{\mathrm{Lip}(h)^3}\|y(0) - y^\star\|^2\,(2 + 4K/3).$$

After rearranging the terms, we obtain

$$\frac{\alpha\mathrm{Lip}(h)}{\|y(0) - y^\star\|}\|w_\alpha(T) - \bar{w}_\alpha(T)\| \leq \frac{K^2}{\alpha}\frac{\mathrm{Lip}(Dh)}{\mathrm{Lip}(h)^2}\|y(0) - y^\star\|\,(2 + 4K/3)$$

Note that this bound is arranged so that both sides of the inequality are dimensionless, in the sense that they would not change under a simple rescaling of either the norm on $\mathcal{F}$ or on $\mathbb{R}^p$. The left-hand side should be understood as the relative difference between the non-linear and the linearized dynamics, while the right-hand side involves the *scale* of Section 1.2.

## B.3 Proof of Theorem 2.4 (over-parameterized case)

Consider the radius $r_0 := \sigma_{\min}/(2\mathrm{Lip}(Dh))$. By smoothness of $h$, it holds $\Sigma(w) \succeq \sigma_{\min}^2 \mathrm{Id}/4$ as long as $\|w - w_0\| < r_0$. Thus Lemma B.1 below guarantees that $y(t)$ converges linearly, up to time $T := \inf\{t \geq 0 \,;\, \|w_\alpha(t) - w_0\| > r_0\}$. It only remains to find conditions on $\alpha$ so that $T = +\infty$. The variation of the parameters $w_\alpha(t)$ can be bounded for $0 \leq t \leq T$ as

$$\|w_\alpha'(t)\| \leq \frac{1}{\alpha}\|Dh(w_\alpha(t))\|\|\nabla R(y(t))\| \leq \frac{2M}{\alpha}\|Dh(w_0)\|\|y(t) - y^*\|.$$

By Lemma B.1, it follows that for $0 \leq t \leq T$,

$$\|w_\alpha(t) - w_0\| \leq \frac{2M^{3/2}}{\alpha m}\|Dh(w_0)\|\|y(0) - y^*\|\int_0^t e^{-(m\sigma_{\min}^2/4)s}ds$$

$$\leq \frac{8\kappa^{3/2}}{\alpha\sigma_{\min}^2}\|Dh(w_0)\|\|y(0) - y^*\|.$$

This quantity is smaller than $r_0$, and thus $T = \infty$, if $\|y(0) - y^*\| \leq 2\alpha C_0$. This is in particular guaranteed by the conditions on $h(w_0)$ and $\alpha$ in the theorem.

When $h(w_0) = 0$, the previous bound also implies the "laziness" property $\sup_{t \geq 0}\|w_\alpha(t) - w_0\| = O(1/\alpha)$ since in that case $y(0)$ does not depend on $\alpha$. For the comparison with the tangent gradient flow, the first bound is obtained by applying the stability Lemma B.2, and noticing that the quantity denoted by $K$ in that lemma is in $O(1/\alpha)$ thanks to the previous bound on $\|w_\alpha(t) - w_0\|$. For the last bound, we compute the integral over $[0, +\infty)$ of the bound

$$\alpha\|w_\alpha'(t) - \bar{w}_\alpha'(t)\| = \|Dh(w_\alpha(t))^\intercal\nabla R(y(t)) - Dh(w_0)^\intercal\nabla R(\bar{y}(t))\|$$

$$\leq \|Dh(w_\alpha(t)) - Dh(w_0)\|\|\nabla R(y(t))\| + \|Dh(w_0)\|\|\nabla R(y(t)) - \nabla R(\bar{y}(t))\|.$$

It is easy to see from the derivations above that the integral of the first term is in $O(1/\alpha)$. For the second term, we define $t_0 := 4\log\alpha/(\mu\sigma_{\min}^2)$ and on $[0, t_0]$ we use the smoothness bound

$$\|\nabla R(y(t)) - \nabla R(\bar{y}(t))\| \leq M\|y(t) - \bar{y}(t)\|$$

which integral over $[0, t_0]$ is in $O(\log\alpha/\alpha)$, while on $[t_0, +\infty)$ we use the crude bound

$$\|\nabla R(y(t)) - \nabla R(\bar{y}(t))\| \leq \|\nabla R(y(t))\| + \|\nabla R(\bar{y}(t))\|$$

which integral over $[t_0, +\infty)$ is in $O(1/\alpha)$ thanks to the definition of $t_0$ and the exponential decrease of $\nabla R$ along both trajectories. This is sufficient to conclude. As a side note, we remark that the assumption that $Dh$ is globally Lipschitz could be avoided by considering the more technical definition

$$\mathrm{Lip}(Dh) \coloneqq \inf \left\{ L > 0 \ ; \ Dh \text{ is } L\text{-Lipschitz on a ball centered at } w_0 \text{ of radius } \frac{\sigma_{\min}}{2L} \right\} > 0,$$

because then the path $w_\alpha(t)$ never escapes the ball of radius $\frac{\sigma_{\min}}{2L}$ around $w_0$ for $\alpha > \|y^*\|/C_0$.

**Lemma B.1** (Strongly-convex gradient flow in a time-dependent metric). *Let $F : \mathcal{F} \to \mathbb{R}$ be a $m$-strongly-convex function with $M$-Lipschitz continuous gradient and with global minimizer $y^*$ and let $\Sigma(t) : \mathcal{F} \to \mathcal{F}$ be a time dependent continuous self-adjoint linear operator with eigenvalues lower bounded by $\lambda > 0$ for $0 \le t \le T$. Then solutions on $[0, T]$ to the differential equation*

$$y'(t) = -\Sigma(t)\nabla F(y(t)),$$

*satisfy, for $0 \le t \le T$,*

$$\|y(t) - y^*\| \le (M/m)^{1/2}\|y(0) - y^*\| \exp\left(-m\lambda t\right).$$

*Proof.* By strong convexity, it holds $\bar{F}(y) \coloneqq F(y) - F(y^*) \le \frac{1}{2m}\|\nabla F(y)\|^2$. It follows

$$\frac{d}{dt}\bar{F}(y(t)) = -\nabla F(y(t))^\intercal \Sigma(t) \nabla F(y(t)) \le -\lambda\|\nabla F(y(t))\|^2 \le -2m\lambda \bar{F}(y),$$

and thus $\bar{F}(y(t)) \le \exp\left(-2m\lambda\right) \bar{F}(y(0))$ by Grönwall's Lemma. We now use the strong convexity inequality $\|y - y^*\|^2 \le \frac{2}{m}\bar{F}(y)$ in the left-hand side and the smoothness inequality $\bar{F}(y) \le \frac{1}{2}M\|y - y^*\|^2$ in the right-hand side. This yields $\|y(t) - y^*\|^2 \le \frac{M}{m} \exp\left(-2m\lambda\right) \|y(0) - y^*\|^2$. $\qquad\square$

## B.4 Stability Lemma

The following stability lemma is at the basis of the equivalence between lazy training and linearized model training in Theorem 2.4. We limit ourselves to a rough estimate sufficient for our purposes.

**Lemma B.2.** *Let $R : \mathcal{F} \to \mathbb{R}_+$ be a $m$-strongly convex function and let $\Sigma(t)$ be a time dependent positive definite operator on $\mathcal{F}$ such that $\Sigma(t) \succeq \lambda\mathrm{Id}$ for $t \ge 0$. Consider the paths $y(t)$ and $\bar{y}(t)$ on $\mathcal{F}$ that solve for $t \ge 0$,*

$$y'(t) = -\Sigma(t)\nabla R(y(t)) \qquad and \qquad \bar{y}'(t) = -\Sigma(0)\nabla R(\bar{y}(t)).$$

*Defining $K \coloneqq \sup_{t \ge 0} \|(\Sigma(t) - \Sigma(0))\nabla R(y(t))\|$, it holds for $t \ge 0$,*

$$\|y(t) - \bar{y}(t)\| \le \frac{K\|\Sigma(0)\|^{1/2}}{\lambda^{3/2}m}.$$

*Proof.* Let $\Sigma_0^{1/2}$ be the positive definite square root of $\Sigma(0)$, let $z(t) = \Sigma_0^{-1/2}y(t)$, $\bar{z}(t) = \Sigma_0^{-1/2}\bar{y}(t)$ and let $h : \mathbb{R}_+ \to \mathbb{R}_+$ be the function defined as $h(t) = \frac{1}{2}\|z(t) - \bar{z}(t)\|^2$. It holds

$$
\begin{aligned}
h'(t) &= \langle z'(t) - \bar{z}'(t), z(t) - \bar{z}(t)\rangle \\
&= -\langle \Sigma_0^{-1/2}\Sigma(t)\nabla R(\Sigma_0^{1/2}z(t)) - \Sigma_0^{1/2}\nabla R(\Sigma_0^{1/2}\bar{z}(t)), z(t) - \bar{z}(t)\rangle \\
&= -\langle \Sigma_0^{1/2}\nabla R(\Sigma_0^{1/2}z(t)) - \Sigma_0^{1/2}\nabla R(\Sigma_0^{1/2}\bar{z}(t)), z(t) - \bar{z}(t)\rangle && (A(t)) \\
&\quad - \langle \Sigma_0^{-1/2}(\Sigma(t) - \Sigma(0))\nabla R(\Sigma_0^{1/2}z(t)), z(t) - \bar{z}(t)\rangle. && (B(t))
\end{aligned}
$$

Since the function $z \mapsto R(\Sigma_0^{1/2}z)$ is $\lambda m$-strongly convex, one has that $A(t) \le -2\lambda m h(t)$. Using the quantity $K$ introduced in the statement, one has also $\|B(t)\| \le K\|z(t) - \bar{z}(t)\|/\sqrt{\lambda} = K\sqrt{2h(t)/\lambda}$. Summing these two terms yields the bound

$$h'(t) \le K\sqrt{2h(t)/\lambda} - 2\lambda m h(t).$$

The right-hand side is a concave function of $h(t)$ which is nonnegative for $h(t) \in [0, K^2/(2\lambda^3 m^2)]$ and negative for higher values of $h(t)$. Since $h(0) = 0$ it follows that for all $t \ge 0$, one has $h(t) \le K^2/(2\lambda^3 \mu^2)$ and the result follows since $\|y(t) - \bar{y}(t)\| \le \|\Sigma(0)\|^{1/2}\sqrt{2h(t)}$. $\qquad\square$

Figure 5: There is a small neighborhood $\mathcal{W}_0 \subset \mathbb{R}^p$ of the initialization $w_0$, which image by $h$ is a differentiable manifold in $\mathcal{F}$. In the lazy regime, the optimization paths (both in $\mathcal{W}$ and in $\mathcal{F}$) for the non-linear model $h$ (dashed gray paths) are close to those of the linearized model $\bar{h}$ (dashed black paths) until convergence or stopping time (Section 2). This figure illustrates the under-parameterized case where $p < \dim(\mathcal{F})$.

## B.5 Proof of Theorem 2.5 (under-parameterized case)

The setting of this theorem is depicted on Figure 5. By the rank theorem (a result of differential geometry, see [21, Thm. 4.12] or [1] for a statement in separable Hilbert spaces), there exists open sets $\mathcal{W}_0, \bar{\mathcal{W}}_0 \subset \mathbb{R}^p$ and $\mathcal{F}_0, \bar{\mathcal{F}}_0 \subset \mathcal{F}$ and diffeomorphisms $\varphi : \mathcal{W}_0 \to \bar{\mathcal{W}}_0$ and $\psi : \mathcal{F}_0 \to \bar{\mathcal{F}}_0$ such that $\varphi(w_0) = 0$, $\psi(h(w_0)) = 0$ and $\psi \circ h \circ \varphi^{-1} = \pi_r$, where $\pi_r$ is the map that writes, in suitable bases, $(x_1, \dots, x_p) \mapsto (x_1, \dots, x_r, 0, \dots)$. Up to restricting these domains, we may assume that $\bar{\mathcal{F}}_0$ is convex. We also denote by $\Pi_r$ the $r$-dimensional hyperplan in $\mathcal{F}$ that is spanned by the first $r$ vectors of the basis. The situation is is summarized in the following commutative diagram:

$$
\begin{array}{ccc}
\mathcal{W}_0 & \xrightarrow{\ h\ } & \mathcal{F}_0 \\
{\scriptstyle \varphi}\downarrow & & \downarrow{\scriptstyle \psi} \\
\bar{\mathcal{W}}_0 & \xrightarrow[\ \pi_r\ ]{} & \bar{\mathcal{F}}_0
\end{array}
$$

In the rest of the proof, we denote by $C > 0$ any quantity that depends on $m$, $M$ and Lipschitz smoothness constants of $h, \psi, \varphi, \psi^{-1}, \varphi^{-1}$, but not on $\alpha$. Although we do not do so, this could be translated into explicit constants that depends on the smoothness of $h$ and $R$, on the strong convexity constant of $R$ and on the smallest positive singular value of $Dh(w_0)$ using quantitative versions of the rank theorem [5, Thm. 2.7].

**Step 1.** Our proof is along the same lines as that of Theorem 2.4, but performed in $\Pi_r$ which can be thought of as a straighten up version of $h(\mathcal{W}_0)$. Consider the function $G_\alpha$ defined for $g \in \bar{\mathcal{F}}_0$ as $G_\alpha(g) = R(\alpha \psi^{-1}(g))/\alpha^2$. The gradient and Hessian of $G_\alpha$ satisfy, for $v_1, v_2 \in \mathbb{R}^p$,

$$
\nabla G_\alpha(g) = \frac{1}{\alpha}(D\psi(g)^{-1})^{\mathsf{T}}\nabla R(\alpha\psi^{-1}(g)),
$$

$$
D^2 G_\alpha(g)(v_1, v_2) = v_1^{\mathsf{T}}(D\psi(g)^{-1})^{\mathsf{T}}\nabla^2 R(\alpha\psi^{-1}(g))D\psi(g)^{-1}v_2
$$
$$
+ \frac{1}{\alpha}D^2\psi(g)^{-1}(v_1, v_2)^{\mathsf{T}}\nabla R(\alpha\psi^{-1}(g)).
$$

The second order derivative of $G_\alpha$ is the sum of a first term with eigenvalues in an interval $[C^{-1}, C]$, and a second term that goes to 0 as $\alpha$ increases. It follows that $G_\alpha$ is smooth and strongly convex for $\alpha$ large enough. Note that if $R$ or $\psi^{-1}$ are not twice continuously differentiable, then the Hessian computations should be understood in the distributional sense (this is sufficient because the functions involved are Lipschitz smooth). Also, let $g^*$ be a minimizer of the lower-semicontinuous closure of

$G_\alpha$ on the closure of $\bar{\mathcal{F}}_0$. By strong convexity of $R$ and our assumptions, it holds

$$\|g^*\|^2 \le \frac{2}{m}(G_\alpha(0) - G_\alpha(g^*)) \le \frac{2R(0)}{\alpha^2 m},$$

so $g^*$ is in the interior of $\bar{\mathcal{F}}_0$ for $\alpha$ large enough and is then the unique minimizer of $G_\alpha$.

**Step 2.** Now consider $T := \inf\{t \ge 0 \; ; \; w_\alpha(t) \notin \mathcal{W}_0\}$. For $t \in [0, T)$, the trajectory $w_\alpha(t)$ of the gradient flow (4) has "mirror" trajectories in the four spaces in the diagram above. Let us look more particularly at $g(t) := \pi_r \circ \varphi(w_\alpha(t)) = \psi \circ h(w_\alpha(t))$ for $t < T$. In the following computation, we write $D\varphi$ for the value of the differential at the corresponding point of the dynamic $D\varphi(w_\alpha(t))$ (and similarly for other differentials). By noticing that $Dh = D\psi^{-1} D\pi_r D\varphi$, we have

$$g'(t) = -\frac{1}{\alpha} D\psi Dh Dh^{\mathsf{T}} \nabla R(\alpha \psi^{-1}(g(t)))$$
$$= -\frac{1}{\alpha} D\pi_r D\varphi D\varphi^{\mathsf{T}} D\pi_r^{\mathsf{T}} (D\psi^{-1})^{\mathsf{T}} \nabla R(\alpha \psi^{-1}(g(t))).$$

so $g(t)$ remains in $\Pi_r$. Also, the first $r \times r$ block of $D\pi_r D\varphi D\varphi^{\mathsf{T}} D\pi_r^{\mathsf{T}}$ is positive definite on $\Pi_r$, with a positive lower bound (up to taking $\mathcal{W}_0$ and $\mathcal{F}_0$ smaller if necessary). Thus by Lemma B.1, there are constants $C_1, C_2 > 0$ independent of $\alpha$ such that, for $t \in [0, T)$, $\|g(t) - g^*\| \le C_1 \|g(0) - g^*\| \exp(-C_2 t)$.

**Step 3.** Now we want to show that $T = +\infty$ for $\alpha$ large enough. It holds

$$w'(t) = -\frac{1}{\alpha} Dh^{\mathsf{T}} \nabla R(\alpha h(w_\alpha(t)) = D\varphi^{\mathsf{T}} D\pi_r^{\mathsf{T}} \nabla G_\alpha(g(t))$$

and, by Lipschitz-smoothness of $G_\alpha$ (Step 1), $\|\nabla G_\alpha(g(t))\| \le \frac{C}{\alpha} \|g(t) - g^*\|$ hence

$$\|w_\alpha(t) - w_0\| \le \frac{C}{\alpha} \int_0^t \exp(-C_2 s) ds \le \frac{C}{\alpha C_2}.$$

Thus, by choosing $\alpha$ large enough, one has $w_\alpha(t) \in \mathcal{W}_0$ for all $t \ge 0$, so $T = \infty$ and the theorem follows.

## C Experimental details and additional results

### C.1 Many neurons dynamics visualized

The setting of Figure 6 is the same as for panels (a)-(b) in Figure 1 except that $m = 200, n = 200$: it allows to visualize behavior of the training dynamics for a larger number of neurons. Symmetrized initialization to set $f(w_0, \cdot) = 0$ was used on panel (c) but not on panel (b), where we see that the neurons need to move slightly more in order to compensate for the non-zero initialization. As on Figure 1, we observe a good behavior in the non-lazy regime for small $\tau$.

(a) Non-lazy training ($\tau = 0.1$)   (b) Lazy ($\tau = 2$, not symmetrized)   (c) Lazy ($\tau = 2$, symmetrized)

Figure 6: Training a two-layer ReLU neural network initialized with normal random weights of variance $\tau^2$, as in Figure 1, but with more neurons. In this 2-homogeneous setting, changing $\tau^2$ is equivalent to changing $\alpha$ by the same amount so lazy training occurs for large $\tau$.

## C.2 Stability of activations

We define here the "stability of activations" mentioned in Section 3.2. We consider a ReLU layer $\ell$ of size $n_\ell$ in a neural network and the test input data $(x_i)_{i=1}^N$ (the test images of CIFAR10 in our case). We call $z_{ij}(T) \in \mathbb{R}$ the value of the pre-activation (i.e. the value that goes through the ReLU function as an input) of index $j$ on the data sample $i$, obtained with the parameters of the network at epoch $T$. The "stability of activations" for this layer is defined as $s_\ell := \frac{Q}{n_\ell \times N}$ where $L$ is the number of ReLU layers, $Q$ is the number of indices $(i, j)$ that satisfy $\operatorname{sign}(z_{ij}(T_{last})) = \operatorname{sign}(z_{ij}(T_{init}))$ for $i \in \{1, \ldots, B\}$ and $j \in \{1, \ldots, n_\ell\}$, where $T_{init}$ refers to initialization and $T_{last}$ to the end of training. The quantity that we report on Figure 3(a) is the average of $s_\ell$ over all ReLU layers of the VGG-11 network, for various values of $\alpha$.

## C.3 Spectrum of the tangent kernel

In the setting of Figure 3(a), we want to understand why the linearized model (that is, trained for large $\alpha$) could not reach low training accuracies in spite of being highly over-parameterized. Figure 7(a) shows the train and test losses after 70 epochs where we see that the training loss is far from 0 for all $\alpha \geq 10$. We report on Figure 7(b) the normalized and sorted eigenvalues $\sigma_i^2$ of the tangent kernel $Dh(w_0)Dh(w_0)^\intercal$ (notice the log-log scale) evaluated for two distinct input data sets $(x_i)_{i=1}^n$ of size $n = 500$: (i) images randomly sampled from the training set of CIFAR10 and (ii) images with uniform random pixel values. Since there are 10 output channels, the corresponding space $\mathcal{F}$ has $10 \times 500$ dimensions. We observe that there is a gap of 1 order of magnitude between the $0.2\%$ largest eigenvalues and the remaining ones—which causes the ill conditionning—and then a decrease of order approximately $O(1/i)$. We observe a similar pattern with the CIFAR10 inputs and completely random inputs, which suggests that this conditioning is intrinsic to the linearized VGG model. Note that modifying the neural network architecture to improve this conditioning, or using optimization methods that are better adapted to ill-conditionned models, is beyond the scope of the present paper.

(a)  (b)

Figure 7: (a) End-of training train and test loss. (b) Spectrum of the tangent kernel $Dh(w_0)Dh(w_0)^\intercal$ for the VGG11 model on two data sets.

## Footnotes

[8] We have omitted the bias/intercept, which is recovered by fixing the last coordinate of $x$ to 1.

[9]Since the definition of gradients depends on the choice of a metric, this scaling is not of intrinsic importance. Rather, it reflects that we work with the Euclidean metric on $\mathbb{R}^p$. The choice of scaling however becomes important when dealing with training (see also discussion in Section 1.2).