[Reviews · NeurIPS 2019]

Reviewer 1



This work provides a unified framework called lazy training to explain some recent success in deep learning theory. In general, it shows that by proper scaling, many real world machine learning applications including two-layer neural networks enjoys properties of lazy training which makes them easier for training. Experiments backup their theory. Overall, this is a good submission and I recommend an accept for this paper. My comments are listed as follows. - It seems that this paper considers both empirical loss and population loss in a general loss function. I suggest the authors to highlight this in their problem setting. #################################### I have read all the reviews and the authors' response.

Reviewer 2



The paper provided some interesting understanding, but is not significant enough to explain interesting issues in deep learning. The paper showed that lazy training can be caused by parameter scaling, not special to overparameterization of neural networks. What does this tell us about the overparameterized neural networks? Does this result imply that lazy regime of overparameterized neural networks is necessarily due to parameter scaling? If not, lazy regime of overparameterized neural networks cannot be explained simply by parameter scaling. I would like to understand the logic of the paper here. What exactly does the paper want to convey? The paper provided experiments to demonstrate that lazy training does not necessarily yield good performance. This is a good observation. However, beyond this, does this tell us anything about overparameterized neural networks? I feel this does not imply that lazy training regime that overparameterized neural networks enter does not provide good performance. I found it is more meaningful to compare the lazy training of overparameterized neural networks and lazying training by large scaling parameter of underparameterized neural networks. I wonder if any of the experiments in the paper can imply any property of such a comparison. ---------- After authors' response The authors answered my questions satisfactorily. I appreciate their efforts into running extra experiments to provide further understanding. Thus, I improve my score to 6.

Reviewer 3



The paper builds on the existing idea of lazy training. where the authors give new insights using the idea about the existance of an implicit scale that controls this phenomenon. This is an interesting idea, nevertheless it feels that they should expand more on this. Technically, the paper is not strong, it feels more like an experimental paper. The idea is novel, I am not sure about the importance of it at this point. ================================================== After rebuttal: I am still not convinced about the significance of this contribution (which is not technical for sure). I keep my score as it is.

[Author Response · NeurIPS 2019]

# Response to reviewers for the paper: "On Lazy Training in Differentiable Programming"

We thank the reviewers for their comments and suggestions. Hereafter, we list reviewers' (sometimes paraphrased) comments (**C**) followed by our responses (**R**). Each answer will translate into a clarification in the final version.

Reviewer #2 and #3 felt that our message was lacking clarity. We would like to re-emphasize that our paper is foremost a reaction to a series of papers (some of them published in the main machine learning venues) with strong claims about optimization of over-parameterized neural networks. We point to the fact that these results are due to a default of normalization, which yields a degenerate "lazy" limit that does not describe well the behavior of competitive over-parameterized models. Along the way, we study lazy training in detail, because it is an interesting novel implicit bias phenomenon in non-convex optimization.

## Response to reviewer #1's comments

- (**C**) *It seems that this paper considers both empirical loss and population loss.* (**R**) Our statements indeed apply to any setting where one wants to fit a non-linear model using a convex loss without regularization. In the experiments, we minimize the population loss in Fig 2.b and the empirical loss everywhere else. We will clarify this.

- (**C**) *The authors should provide analysis about the generalization behavior about two-layer neural networks.* (**R**) We would like to maintain our focus on the optimization aspect, which is where lazy training is new. From a statistical aspect, the model behaves like its linearization (see Prop. A.1, up to a $\tilde{O}(1/\alpha)$ error), in this case a random feature model (Sec. A.2). We will add more pointers to their statistical analysis, from the existing literature (e.g. F. Bach, *On the Equivalence between Kernel Quadrature Rules and Random Feature Expansions*, JMLR 2017) and also from follow up work from other authors[1].

## Response to reviewer #2's comments

- (**C**) *What does this analysis tell about over-parameterized neural networks?* We reiterate that the lazy behavior of over-parameterized two layer neural networks, discussed in a series of paper (e.g., [11, 21, 10, 2, 3, 36]), is due to an implicit choice of degenerate scaling (cf. L81-90 in the main paper, often $\alpha(m) = 1/\sqrt{m}$ in these works). Instead, we show that with the scaling $\alpha(m) = 1/m$, over-parameterization is unrelated to laziness. We also show that the lazy regime leads to poorer performances, and thus should be avoided (see e.g. Fig. 2.(a)).

- (**C**) *Can any of the experiments in the paper help to compare the behavior of lazy training in over-parameterized vs not models?* This is a pertinent comment. We have conducted additional experiments that have shown that even very wide neural networks perform poorly in the lazy regime. We started from a standard CNN (VGG) on CIFAR and widened each layer by a factor 8, implying that the number of parameters was multiplied by roughly $8^2$. Trained in the lazy regime, we obtained the poor performance of $61.7\%$ test accuracy against $89.7\%$ for its non-lazy counterpart, which is consistant with our claims. We will add this result in the final version of the paper.

## Response to reviewer #3's comments

- (**C**) *The paper builds on the existing idea of lazy training.* (**R**) The term "lazy training" is introduced in this paper. Previous works have proved that over-parameterized neural networks could have a lazy behavior, but we are the first to put forward the phenomenon at play (degenerate scaling), to show its generality (beyond over-parameterization and beyond neural networks), its drawbacks (features are not learnt) and how to avoid it (through scaling or initialization).

- (**C**) *Technically, the paper is not strong, it feels more like an experimental paper.* (**R**) We agree that once the notion of scale is isolated, the theoretical results are almost elementary (to the exception of Thm. 2.5, as noted by reviewer #1). This is actually our goal to put simplicity forward and we believe this should rather be considered as a strength.

- (**C**) *The idea is interesting but I am not sure about its importance.* (**R**) To us, this paper is important for two reasons:
    - it mitigates the claim of the series of work on neural networks optimization, which is needed for practitioners to not search for lazy training, and as well for theory of over-parameterization to be explored in other directions;
    - the notion of lazy training explains the behavior of a large class of models in certain regimes of hyper-parameters. Although in this paper, to fix ideas, the scale is explicitly represented by $\alpha$, it translates in practice to the variance of the initialization, the number of neurons or the normalization of the labels, which practitioners have to deal with when defining a model.

## Footnotes

[1]Which we do not cite to preserve anonymity.


[Meta-Review · NeurIPS 2019]

The paper received no strongly positive scores 6/5/6 at the beginning, with acceptable confidences 4/4/3. The author rebuttal successfully convinced reviewers #1 and #2, who both raised their scores by 1. But reviewer #3 was still not fully convinced so s/he maintained her/his score. So the new scores are 7/6/6. The AC browsed through the paper and rebuttal, and found that the paper does provide good insight into the performance of DNNs, although the DNN models assumed are still far from those in real use. So the AC recommended accepting as poster.